

# The farther, the better? The effect of attentional focus distance on motor performance: a systematic review and meta-analysis

Le Zang, Wei Guo and Biye Wang

College of Physical Education, Yangzhou University, Yangzhou, China

## ABSTRACT

**Background.** Motor skill performance is influenced by attentional focus, and recent studies suggest that external focus, particularly a more distal external focus, may enhance performance more than an internal or a proximal external focus. However, the optimal distance for external focus and the influence of expertise level remain unclear. This systematic review and meta-analysis aimed to evaluate the effect of attentional focus distance on motor performance across various tasks and expertise levels.

**Methods.** This systematic review and meta-analysis followed established guidelines (PRISMA) and was registered with the International Prospective Register of Systematic Reviews (CRD42024595116). Comprehensive literature searches were performed in PubMed, Web of Science, Embase, and Cochrane Library databases. Data extraction followed the PICOS framework, and analyses were conducted using Stata and Review Manager.

**Results.** A total of 20 randomized controlled trials with 497 participants were included. The results indicated that a distal external focus significantly improved motor performance compared to both external proximal focus (standardized mean difference, S = 0.3, 95% CI [0.07–0.53], $p = 0.01$) and internal focus (SMD = 0.59, 95% CI [0.14–1.05], $p = 0.01$). However, no significant difference was observed between the distal and proximal external focus in novices. Subgroup analyses showed that skill level significantly moderated the effectiveness of attentional focus.

**Conclusions.** A distal external focus improves motor performance in experienced although its effectiveness in novices requires further study. Future research should explore the mechanisms underlying these effects and skill-specific guidelines for optimal attentional focus distance.

## INTRODUCTION

Attentional focus, a critical concept in motor performance research, refers to how performers allocate their attentional resources to the instructional cues provided throughout teaching. It is categorized into internal focus (*i.e.,* focusing on body movement) and external focus (*i.e.,* focusing on an object, movement effector, or target)

Corresponding author
Biye Wang, wangbiye@yzu.edu.cn

(*Wulf, 2013*). A recent meta-analysis indicates that external focus instructions lead to superior motor performance compared to internal focus, regardless of participants' skill levels or task demands (*Chua et al., 2021*). For instance, in a golf putting task, experienced individuals performed significantly better when focusing on the target than on their body movements (*Wulf & Su, 2007*). The same result was also observed in a shooting task with a more open environment showed that novices achieved better shooting performance and maintained longer fixation times when focusing on the basket rather than on their body movements (*Asadi et al., 2023*). Further evidence indicates that external focus enhance performance across various activities, including balance tasks, throwing sports, ball games, and electromyography (EMG) measurements (*Wulf, McNevin & Shea, 2001*; *Emanuel, Jarus & Bart, 2008*; *Chiviacowsky, Wulf & Wally, 2010*; *Perreault & French, 2015*).

To explain the robust effectiveness of external focus, Wulf introduced the constrained action hypothesis, which assumed that focus on one's own movements disrupts the coordination of motor system, impairing automatic control processes and leading to movement dysregulation (*Wulf, McNevin & Shea, 2001*). By contrast, an external focus engages unconscious, rapid, and reflexive processes that promote a more automatic control mode (*Wulf, 2013*). Further support for the constrained action hypothesis has come from studies using probe-response time task, dual-task paradigm, and vector coding technique (*Kal, Van der Kamp & Houdijk, 2013*; *Vidal et al., 2018*; *Allingham & Wollner, 2022*).

While the benefits of external focus are relatively consistent, not all external foci yield equivalent effects on motor performance. The efficacy of external focus on motor performance may depend on focus distance (*McNevin, Shea & Wulf, 2003*). Specifically, external focus can be directed proximally (near the body) or distally (toward a distant target). Previous research generally indicates that distal external focus yields better performance than proximal external focus. For instance, in dart throwing tasks, focusing on the center of the target results in significantly higher scores than focusing on the trajectory of the dart (*Simpson et al., 2022*). A previous study showed that distal external focus may help learners concentrate their attention on task-related goals, thereby reducing the cognitive load associated with consciously controlling the action itself (*Goncalves Pereira-Junior & Gatinho Bonuzzi, 2022*). This result is also consistent with the findings of *McNevin, Shea & Wulf (2003)*, who proposed that distal external focus promotes more automatic motor control by encouraging people to focus on the effects of movements further away from the body, providing further support for the constrained action hypothesis (*McNevin, Shea & Wulf, 2003*; *Wulf, 2013*). Importantly, focusing on a distal external target does not mean separating the participant's actions from the results. Instead, it emphasizes the potential benefits of maintaining perceptual-motor coupling, which allows performers to better connect their actions with the expected effects. This finding is similarly supported by *Niznikowski et al. (2022)* and *Bull et al. (2023)*, who found that both novice table tennis players and experienced golfers performed better when instructed with a distal external focus, compared to either proximal external focus or internal focus.

The distance effect of external attentional focus has been confirmed by multiple studies (*Wulf, 2013*; *Chua et al., 2021*). Nevertheless, studies involving participants of varying skill

levels have revealed inconsistent results regarding the distance effect of attentional focus. Specifically, distal external focus is not universally optimal for all performer levels. Recent research has found that proximal external focus may, in some cases, be more beneficial for novices. For example, in a golf putting task, *Chen et al. (2022)* found that when novices engaged in complex motor skill training, a distal external focus might impair performance because they had not yet developed the corresponding motor programs. In contrast, novices performed best when adopting a proximal external focus. A similar effect was reported for novice dart players executing backhand strokes (*McKay & Wulf, 2012*). *Singh & Wulf (2020)* likewise found that although distal focus enhanced volleyball performance in experienced athletes, proximal external focus proved more beneficial for novices. Similarly, *Roberts & Lawrence (2019)* reported that novices achieved better performance using proximal external focus rather than distal focus during a digitized target task. These findings run counter to the predictions of the constrained action hypothesis, suggesting that more distant attentional focus does not necessarily benefit novices.

OPTIMAL theory is a motor learning framework proposed by *Wulf & Lewthwaite (2016)* that emphasizes optimizing learning performance by integrating intrinsic motivation and attention mechanisms. According to this theory, the mechanism underlying how attentional focus optimizes motor performance is not singular but rather depends on the attentional demands of the task, alongside enhanced expectations and motivational support (*Wulf & Lewthwaite, 2016*). Variations in attentional demands may alter the coupling of goals and actions, as well as dopamine availability, resulting in differences in neural plasticity that may explain these inconsistencies. Another perspective, the hierarchy of action goal hypothesis, posits that the optimal external focus distance is influenced by both the performer's skill level and the complexity of the task (*Stoate & Wulf, 2011*). For novices, a proximal external focus may be advantageous because it avoids the negative effects of an internal focus while providing key information about expected movement patterns. In the early stages of learning, when stable movement representations have not yet been formed, novices typically rely on sensory feedback to adjust their movements. Therefore, focusing on an external focus that is closer to the body in space, such as the trajectory of the club swing, can provide clearer information about the outcome of the movement, thereby strengthening the connection between perception and action. For experienced participants, their movement representations are relatively stable, and a distal external focus can further enhance the automation of movements and prevent interference caused by excessive self-focus.

A review of studies that found inconsistent or contradictory results suggests that methodological differences might also play a role. For example, most randomized controlled trials (RCTs) use a within-group design in which participants perform motor tasks under varying attentional focus instructions in a randomized sequence. In contrast, *Chen et al. (2022)* employed a between-group design. Despite randomization, each design type carries distinct biases. Specifically, all participants in a within-subjects design are required to perform the task in each attentional focus condition, which controls for between-individual variability but may introduce carry-over or order effects. In contrast, the between-subjects design randomly assigned participants to only one condition, thereby minimizing learning

or fatigue effects across conditions, but increasing the risk of between-group variability and having lower statistical power due to smaller sample sizes in each group (*Wulf, 2013*; *Ranganathan, Lee & Krishnan, 2022*). Furthermore, the environment in which the tasks are executed may also contribute to these differences. Unlike the majority of studies that utilize closed-skill tasks, Singh investigated a volleyball passing task in a more open environment, necessitating players to respond to ball trajectories characteristic of open-skill sports (*Singh & Wulf, 2020*). Open skills are executed in unpredictable and dynamic environments (*e.g.*, volleyball or soccer) and require greater attentional resources for perceptual monitoring and motor adjustments. In contrast, closed skills are performed in stable and predictable environments (*e.g.*, dart throwing or golf putting), thereby allowing the execution of pre-programmed movements. Previous research suggests that these types of tasks impose distinct attentional demands, which may influence the effects of attentional focus (*Kelso, 1992*). As these differences may influence the sensitivity and generalizability of the results, analyzing study design and skill type is essential for accurately interpreting the effects of attentional focus distance.

Two previous meta-analyses examined the effects of attentional focus on motor performance. One showed that distal external focus was superior to both proximal external and internal focus across different ages, health status, and skill levels (*Chua et al., 2021*). However, this study only included 10 studies on the distance effect, with high heterogeneity. Another meta-analysis concluded that increasing the distance of external focus of attention improves motor performance, but it included only six relevant studies, predominantly within-subjects design, and did not adequately consider between-subjects designs (*Nicklas et al., 2022*). Moreover, the type of motor tasks has been recognized as a potential moderating factor. Therefore, three subgroups were explored in this meta-analysis: skill type (open *vs.* closed), skill level (novice *vs.* experienced), and experimental design (between-subjects *vs.* within-subjects).

Based on previous research, this meta-analysis includes additional studies and thoroughly examines the effects of within-subject design. The aim of this study is to validate the effect of attentional focus distance on motor performance and to test the constrained action hypothesis and the action-goal hierarchy hypothesis in the light of the results of the meta-analysis.

## MATERIALS AND METHODS

This meta-analysis was conducted in accordance with the guidelines outlined in the Preferred Reporting Items for Systematic Reviews and Meta-Analyses (PRISMA) statement. The review protocol was also registered in the International Prospective Register of Systematic Reviews (PROSPERO, CRD42024595116). The literature search will follow the PRISMA guidelines and risk of bias assessment will be performed using the Cochrane RoB 2 tool.

The audience for this study includes kinesiology researchers, professionals and academics, as well as students and educators in the field. The focus is on the effects of attentional focus at different distances on motor performance. The review covers relevant
publication information, datasets, test methods, and performance assessments, providing a more comprehensive and balanced approach than previous reviews.

## Search strategy

Systematic literature searches were performed in the PubMed, Web of Science, Embase, and Cochrane Library databases, encompassing literature published up to September 2024. The search strategy combined subject-specific and free-text terms, and supplemented by manual searching. Where necessary, references cited within the retrieved articles were traced to identify additional relevant studies. The key search terms included: (''attention focus'' OR ''focus of attention'' OR ''attentional focus'' OR ''internal focus'' OR ''external focus'' OR ''proximal external focus'' OR ''distal external focus'') AND (''motor skill'' OR ''motor skill learning'' OR ''motor performance''). In addition, the reference lists of the retrieved articles were manually examined to identify other relevant studies. Two independent reviewers (L. Zang and W. Guo) did the screening based on titles and abstracts initially. The remaining articles were further screened by full-text assessment. Any disagreements between the two reviewers were resolved through consensus and by discussion with a third author (B. Wang).

## Literature inclusion and exclusion criteria

This study adhered to international guidelines for conducting meta-analyses, utilizing the PICOS framework to select literature and establishing specific inclusion and exclusion criteria for the studies. Studies were included in this meta-analysis if they met the following eligibility criteria: (1) population: physically fit individuals, including adults, children, and adolescents; (2) intervention: Interventions employing attention-focused guidance or feedback at varying distances, including distal, proximal external, and internal attentional focus; (3) comparison: comparison of distal external focus with other focus conditions, including at least one control group (proximal external focus or internal focus); (4) outcome: motor performance scores measured under various focal conditions, including at least one outcome suitable for calculating effect sizes; (5) study design: randomized controlled trial (RCT) design; (6) language: written in English. The following types of studies were excluded: (1) population: studies involving individuals with non-health-related conditions, such as attention deficit disorders; (2) intervention: interventions not employing attention-focused guidance or feedback; (3) comparison: studies that did not compare distal attentional focus with either proximal or internal focus conditions; (4) outcome: studies without relevant data on attentional focus intervention; (5) study design: studies without a RCT design; (6) other criteria: non-interventional studies, reviews and theoretical articles, case reports and protocol papers, unpublished studies, and articles not written in English.

## Literature quality assessment

Prior to data analysis, the Cochrane RoB 2 tool was used to systematically assess the potential risk of bias in the included studies. It contains six items to assess random allocation concealment, allocation scheme concealment, blinding, completeness of outcome data, selective reporting of findings and other sources of bias. In conducting the literature quality

assessment, two researchers categorized the included literature according to three levels of low risk of bias, high risk of bias, and unclear risk of bias. Disagreements were resolved by a third researcher. These assessments were used to inform later interpretation of findings, including sensitivity analyses and heterogeneity evaluations.

## Data extraction

After eliminating duplicates, two researchers independently screened all titles and abstracts based on the inclusion and exclusion criteria. Full texts of the selected studies were then reviewed. Discrepancies during the screening process were resolved through discussion with a third author's opinion until consensus was achieved. However, inter-rater reliability was not formally assessed. For all included studies, data from ten key aspects were recorded and summarized. These aspects included: (1) motor task performed; (2) number of participants; (3) first author and year published; (4) age of the participants; (5) research design; (6) expertise sub-group; (7) kind of attentional focus instruction; (8) outcome indicators; (9) skills types; (10) statistical results comparing attentional focus conditions.

Data on attentional focus tasks were extracted based on comparisons of distal focus *versus* proximal external focus or internal focus, as well as proximal external focus *versus* internal focus alone. Mean values, standard deviations, and participants numbers for each group at both pretest and posttest were extracted. When these values were unavailable, change values in means and standard deviations following intervention or mean difference with 95% confidence interval were extracted instead.

Given that different experimental designs may lead to varying biases and have varying effects on motor performance. The present study included both within- and between-study designs to provide a comprehensive picture of the available evidence on the effects of attentional focus. To account for differences in effect size estimates that may result from study design, we conducted subgroup analyses based on study design and examined heterogeneity to assess its impact.

Participants were categorized into two sub-groups (experienced, novice) based on study descriptions. Participants were assigned to the novice if they had no prior experience with the task, had performed it only a few times, or were described as novices, untrained, or recreationally active college students. Participants labeled as practiced, highly experienced, elite, experts, or members of national squad or professional squad were included into the experienced subgroup.

When a study included multiple tasks assessing motor performance, we aggregated effect sizes into a composite effect size. Specifically, we calculated the composite effect size as the arithmetic mean of the individual standardized mean differencs (SMDs) following the method recommended by *Borenstein et al. (2009)* and calculated the corresponding variance according to the formula. This method takes into account correlations between outcomes.

## Statistical analysis

Review Manager 5.3 and Stata 16.0 were used to analyze the extracted data. Since the extracted data were continuous, SMD and 95% confidence intervals (CIs) were used as

evaluation indexes. The SMD was calculated as the mean of the difference between groups divided by the standard deviation of the post-intervention scores, with thresholds of 0.2, 0.5, and 0.8 representing small, medium, and large effect sizes, respectively (*Borenstein et al., 2010*).

$I^2$ statistic and *Q*-test were used to assess heterogeneity according to the Cochrane Handbook. When the *p*-value of the *Q*-test was less than 0.05 and the $I^2$ value exceeded 50%, significant heterogeneity was indicated, and a random-effects model was applied. Otherwise, a fixed-effects model was used. For significant heterogeneity, subgroup analyses were conducted to identify potential sources.

Sensitivity analysis was performed using Stata 16.0. The analysis consisted of sequential exclusion of the overall results. If the $I^2$ value was ≤ 50% and the *p*-value ≥ 0.1, a fixed-effects model was used. Otherwise, a random-effects model was applied.

Publication bias was assessed using funnel plots and Egger's linear regression test. Statistical significance was set at $p < 0.05$. If asymmetry was observed in the funnel plot or Egger's test indicated potential bias, a nonparametric trim-and-fill analysis was subsequently conducted. This method corrects for potential publication bias by trimming (identifying and removing studies that cause funnel plot asymmetry to estimate a "trimmed" central effect size) and filling (adding "missing" studies to the opposite side of the funnel plot to simulate the influence of unpublished studies and estimate a bias-adjusted overall effect size) (*Duval & Tweedie, 2000*). A substantial reduction in the effect size after the trim-and-fill procedure would indicate the presence of potential publication bias. Conversely, if the effect size remains stable, the influence of publication bias is considered minimal.

## RESULTS

### Study selection and characteristic

After completing the search strategy, a total of 4,259 articles were obtained and identified as qualifications of inclusion criteria. After removing 413 duplicates, 3,846 remained to review. Two reviewers independently read the title and abstract of each included record using Endnote 20 software based on the inclusion and exclusion criteria. As a result, 36 articles were identified for full-test screen.

After full-text screen, sixteen articles were excluded for the following reasons: non-RCT design ($n = 6$), no motor performance outcomes or insufficient date ($n = 9$), and lack of proximal focus groups ($n = 1$). Finally, a total of 20 RCTs were included in the final analyzed (Fig. 1).

Across these 20 included RCTs, the total sample size was 497 participants, the majority of whom were young or middle-aged adults. The experimental group received distal external attentional focus instruction, while the control group was instructed with either proximal external attentional focus or internal attentional focus.

Among the included studies, 16 adopted within-subjects design (*McKay & Wulf, 2012*; *Porter et al., 2013*; *Pelleck & Passmore, 2017*; *Marchant et al., 2018*; *Roberts & Lawrence, 2019*; *Kupper et al., 2020*; *Singh & Wulf, 2020*; *Banks et al., 2020*; *Singh et al., 2022*; *Bull et*

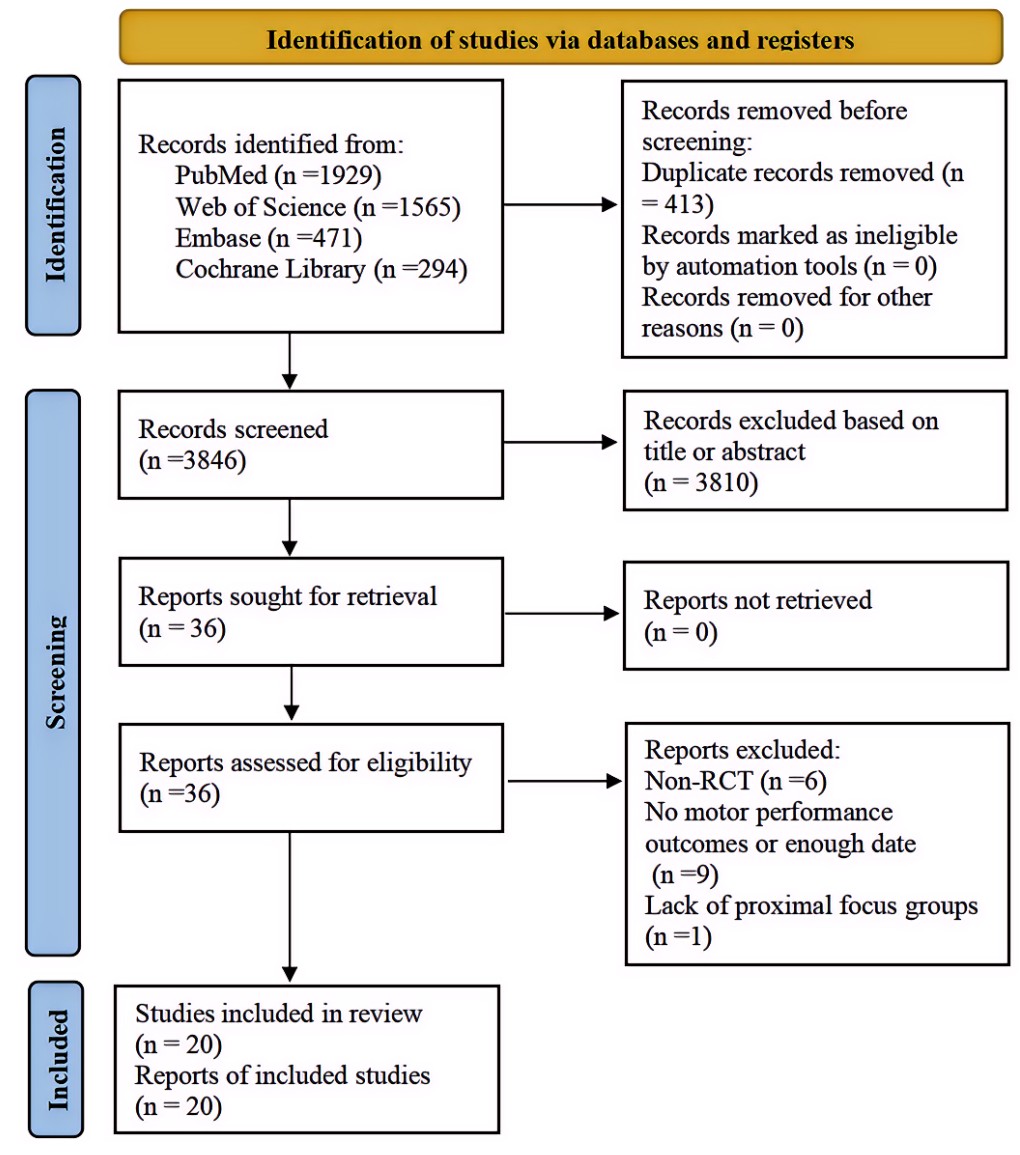

**Figure 1  Selection process for the meta-analysis.**

al., 2023; Peng et al., 2024), and four used between-subject design (Bell & Hardy, 2009; Nagano, Hata & Nagano, 2020; Chen et al., 2022; Niznikowski et al., 2022). Eight of the studies focused on experienced individuals, seven focused on novices, and the other five did not report explicitly on skill levels. Further details are presented in Table 1.

## Quality assessment of included studies

Comprehensive insights into potential bias risks were meticulously illustrated in Figs. S1 and S2, utilizing the Cochrane risk of bias tool for rigorous assessment. All of the included studies provided transparent and comprehensive details about the random sequence generation process. However, for the aspect of allocation concealment, a large portion of

**Table 1** Characteristics of included studies.

| Study | Sample size | Age | Skill level | Design of experiments | Verbal instruction | Task | Skill type | Outcome measure |
|---|---|---|---|---|---|---|---|---|
| *Banks et al. (2020)* | 27 | 41.3 ± 13.1 | Experienced | within-subject | **PEF:** "Think only about the paddle. Use it as well as you can to sprint fast." **DEF:** "Think only about the finish. Imagine arriving as fast as you can." | Rowing | Open | Contact time |
| *Bell & Hardy (2009)* | 33 | 37.1 ± 13.8 | Experienced | between-subject | **INF:** "Focus on the movement of the arms during the swing and specifically keep the wrists hinged" **PEF:** "Focus on keeping the clubface in place during the swing" **DEF:** "Focus on the flight path of the ball after it leaves the club" | Golf putting | Closed | Distance to target |
| *Bull et al. (2023)* | 13 | 35.5 ± 12.0 | Experienced | within-subject | **INF:** "focus on the movement of your hands" **PEF:** "focus on the movement of your bat" **DEF:** "focus on the ball flight of your shot" | Cricket batting | Open | Number of errors |
| *Chen et al. (2022)* | 45 | 22.3 ± 4.7 | Novice | between-subject | **INF:** "Focus on your arm movement" **PEF:** "Focus on the club" **DEF:** "Focus on the target" | Golf putting | Closed | Putt Points |
| *Porter et al. (2013)* | 38 | 20.7 ± 2.2 | Experienced | within-subject | **INF:** "Focus on features of body movement, such as quick knee extensions" **PEF:** "Focus on jumping over the starting line" **DEF:** "Focus on jumping to markers that are 3 m away" | Jumping | Closed | Jump distance |
| Pelleck 2017a (*Pelleck & Passmore, 2017*) | 13 | 32.8 ± 14.4 | Experienced | within-subject | **INF:** "focus on distributing your weight evenly through both feet" | Golf putting | Closed | Radial error |
| Pelleck 2017b (*Pelleck & Passmore, 2017*) | 11 | 33.5 ± 13.2 | Novice | | **PEF:** "focus on your hands gripping the club and the position of your elbows" **DEF:** "focus on the target" | | | |
| *Roberts & Lawrence (2019)* | 15 | 18–21 | – | within-subject | **INF:** "FOCUS on the HAND" **PEF:** "FOCUS on the CURSOR" **DEF:** "FOCUS on the TARGET" | digitized target task | Closed | Radial error |
| Peng 2024a (*Peng et al., 2024*) | 10 | 20 ± 1.3 | Novice | within-subject | **INF:** "When jumping, rapidly extend your hip and knee joints" **PEF:** "When jumping, try to get as high as possible off the ground" **DEF:** "When jumping, try to get as close as possible to the ceiling" | counter movement jump | Closed | Impulse-Momentum Calculus |
| Peng 2024b (*Peng et al., 2024*) | 11 | 20 ± 1.4 | Experienced | | | | | |
| *Porter, Anton & Wu (2012)* | 35 | 22.3 ± 2.5 | – | within-subject | **PEF:** "jump as far past the start line as possible" **DEF:** "jump as close to the cone as possible" | Jumping | Closed | Jump distance |
| *Nagano, Hata & Nagano (2020)* | 20 | 20.5 ± 1.5 | – | between-subject | **PEF:** "Focus on the -20cm target line" **DEF:** "Focus on the +20cm target line" | Jumping | Closed | Jump distance |
| *Marchant et al. (2018)* | 54 | 7.35 ± 1.7 | Novice | within-subject | **INF:** "focus on extending your legs as rapidly as possible" **PEF:** "jump as far past the start line as possible" **DEF:** "jump as close to the cone as possible" | Jumping | Closed | Jump distance |
| Singh 2020a (*Singh & Wulf, 2020*) | 17 | 23.8 ± 6.67 | Novice | within-subject | **PEF:** "concentrate on your platform" | volleyball passing | Open | Accuracy score |
| Singh 2020b (*Singh & Wulf, 2020*) | 12 | 23 ± 3.95 | Experienced | | **DEF:** "concentrate on the bullseye" | | | |

| Study | Sample size | Age | Skill level | Design of experiments | Verbal instruction | Task | Skill type | Outcome measure |
|---|---|---|---|---|---|---|---|---|
| *Singh et al. (2022)* | 20 | $25.2 \pm 4.71$ | Experienced | within-subject | **INF**: "focus on your hand while contacting the ball" **PEF**: "focus on contacting the middle of the ball" **DEF**: "focus on hitting the bullseye" | volleyball serve | Open | Accuracy score |
| *Niznikowski et al. (2022)* | 51 | $22.9 \pm 1.8$ | Novice | between-subject | **INF**: "concentrate on the hand holding the paddle" **PEF**: "concentrate on the ball" **DEF**: "concentrate on targets marked on the tennis table" | Table Tennis Hitting | Open | Accuracy score |
| *McKay & Wulf (2012)* | 36 | 21.3 | Novice | within-subject | **PEF**: "focus on the flight of the dart" **DEF**: "focus on the bull's eye" | Dart throwing | Closed | Accuracy score |
| Kupper 2020a (*Kupper et al., 2020*) | 18 | $23 \pm 2$ | – | within-subject | **PEF**: "Focus on controlling the laser pointer directly attached to your forehead" | SLS Balance Task | Closed | MSEN |
| Kupper 2020b (*Kupper et al., 2020*) | 18 | $25 \pm 4$ | – | | **DEF**: "Focus on controlling the laser attached to the top of your cone" | | | |

**Notes.**

DEF, distal external focus; PEF, proximal external focus; INF, internal focus; SLS, single leg stand; MSEN, multiscale entropy; -, unreported.

the studies omitted reporting or did not have enough information to determine whether allocations were concealed. Given the nature of the interventions, in all included studies, blinding of participants and researchers was considered to be challenging. In all studies, participants were labeled as "high risk" because they were all aware of their specific task and the type of attentional focus. In the context of outcome assessment blinding, eight studies were marked as "unclear" and eleven study was assigned a high risk. Noteworthy, no other assessed domains were identified as "high risk".

## Meta-analysis of attentional focus distance on motor performance
### Distal external focus versus proximal external focus
A total of 20 studies were included in the Meta-analysis comparing distal *versus* proximal external attentional focus. The results of the funnel plot and Egger's test ($p = 0.494$), indicated no significant publication bias (Fig. S3). As illustrated in Fig. 2, significant heterogeneity was detected based on the Cochrane Q statistic and the $I^2$ statistic ($Q_{(19)} = 46.36$, $p < 0.001$, $I^2 = 59\%$), necessitating the use of a random-effects model for the analysis. The results demonstrated that motor performance under the distal external focus condition was superior to that of the proximal external focus condition (SMD = 0.3, 95% CI [0.07–0.53], $p = 0.01$).

### Distal external focus versus internal focus
Eleven studies compared the effect of external focus to internal focus. The funnel plot and Egger's test ($p = 0.191$) showed that there was no significant publication bias in the results, see Fig. S4. Significant heterogeneity was detected, as evidenced by the Cochrane Q statistic and the $I^2$ statistic ($Q_{(11)} = 55.31$, $p < 0.001$, $I^2 = 80.1\%$). Consequently, a random-effects model was employed to analyze the study. As shown in Fig. 3, the meta-analysis revealed that motor performance was significantly enhanced under the distal focus condition compared to internal attentional focus (SMD = 0.59, 95% CI [0.14–1.05], $p = 0.01$).

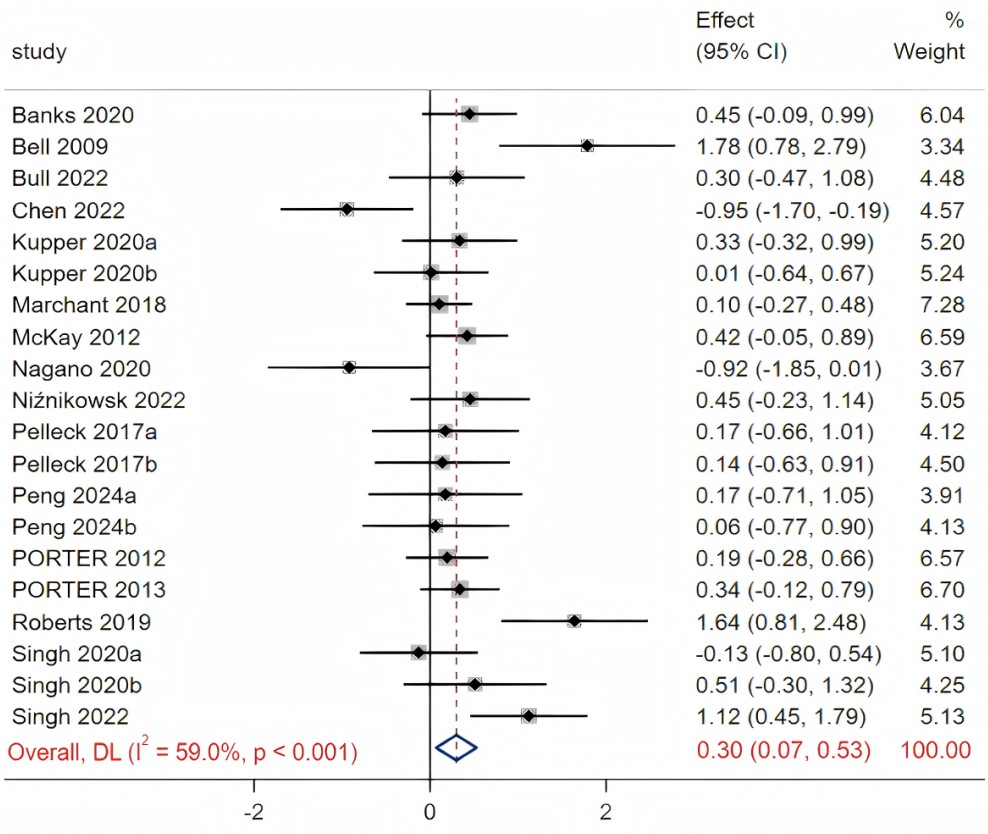

**Figure 2** Forest plot of distal external focus *versus* proximal external focus. Sources: *Banks et al., 2020;* *Bell & Hardy, 2009; Bull et al., 2023; Chen et al., 2022; Porter, Anton & Wu, 2012; Porter et al., 2013; Pelleck & Passmore, 2017; Roberts & Lawrence, 2019; Peng et al., 2024; Nagano, Hata & Nagano, 2020; Marchant et al., 2018; Singh et al., 2022; Singh & Wulf, 2020; Niznikowski et al., 2022; McKay & Wulf, 2012; Kupper et al., 2020.*

## Proximal external focus *versus* internal focus

The initial funnel plot revealed significant asymmetry, and Egger's test indicated significant publication bias ($p = 0.027$). Consequently, we excluded one study exhibiting excessive bias (*Bell & Hardy, 2009*). The revised funnel plot is presented in Fig. S5, indicated no significant publication bias following this exclusion ($p = 0.32$). The heterogeneity test did not reach statistical significance ($Q_{(10)} = 15.97$, $p = 0.1$, $I^2 = 37.4\%$), thus a fixed-effects model was employed for the analysis. Ten studies comparing the effect of proximal external focus to internal focus were included. As shown in Fig. 4, the overall motor performance of the proximal external focus group was significantly better than in the internal focus group ($SMD = 0.32$, 95% CI [0.13–0.51], $p = 0.001$).

## Subgroup analysis

The results of the subgroup analysis, including SMD values, 95% confidence intervals (CIs), and statistical values for the homogeneity test, are presented in Table 2. The
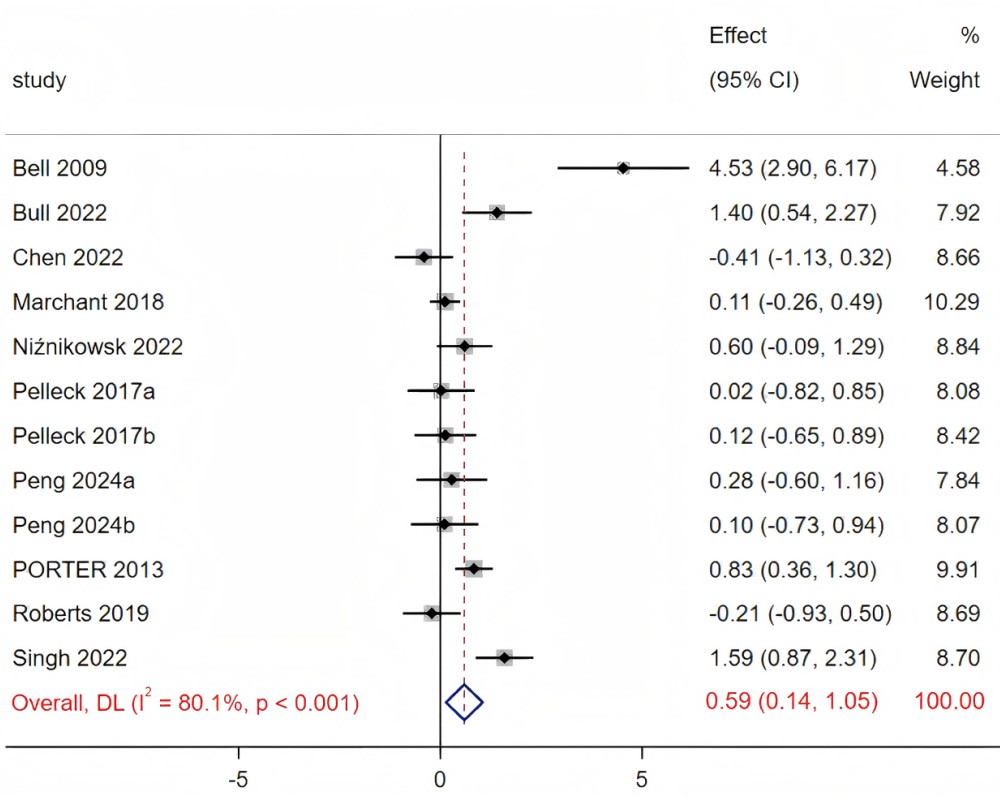

**Figure 3** Forest plot for the effect sizes of the distal external focus compared to the internal focus. Sources: *Bell & Hardy, 2009*; *Bull et al., 2023*; *Chen et al., 2022*; *Porter et al., 2013*; *Pelleck & Passmore, 2017*; *Roberts & Lawrence, 2019*; *Peng et al., 2024*; *Nagano, Hata & Nagano, 2020*; *Marchant et al., 2018*; *Singh et al., 2022*; *Singh & Wulf, 2020*; *Niznikowski et al., 2022*; *McKay & Wulf, 2012*; *Kupper et al., 2020*.

following variables were analyzed: skill level of participants (experienced or novice); design of included studies (within-subject or between-subject) and type of motor skills (open skill or closed skill).

In the comparison of distal and proximal external focus, the heterogeneity test indicated that skill level significantly influenced the effects of attentional focus on motor performance ($Q_{(14)} = 27.32$, $p = 0.02$, $I^2 = 49\%$). Distal focus showed a significant advantage in enhancing motor performance among experienced participants compared to proximal external focus (SMD = 0.5, 95% CI [0.26–0.73], $p < 0.001$). However, the improvement in motor performance among novice did not reach statistical significance (SMD = 0.1, 95% CI [−0.12–0.32], $p > 0.05$). In terms of experimental design, distal focus led to statistically significant improvements in motor performance within the within-group design (SMD = 0.33, 95% CI [0.15–0.50], $p < 0.001$), though no significant effects were observed in the between-group design (SMD = 0.07, 95% CI [−1.05–1.19], $p > 0.05$). Furthermore, distal focus significantly improved motor performance in both open (SMD = 0.59, 95% CI [0.25–0.93], $p < 0.01$) and closed (SMD = 0.2, 95% CI [0.04–0.35], $p < 0.05$) skills.

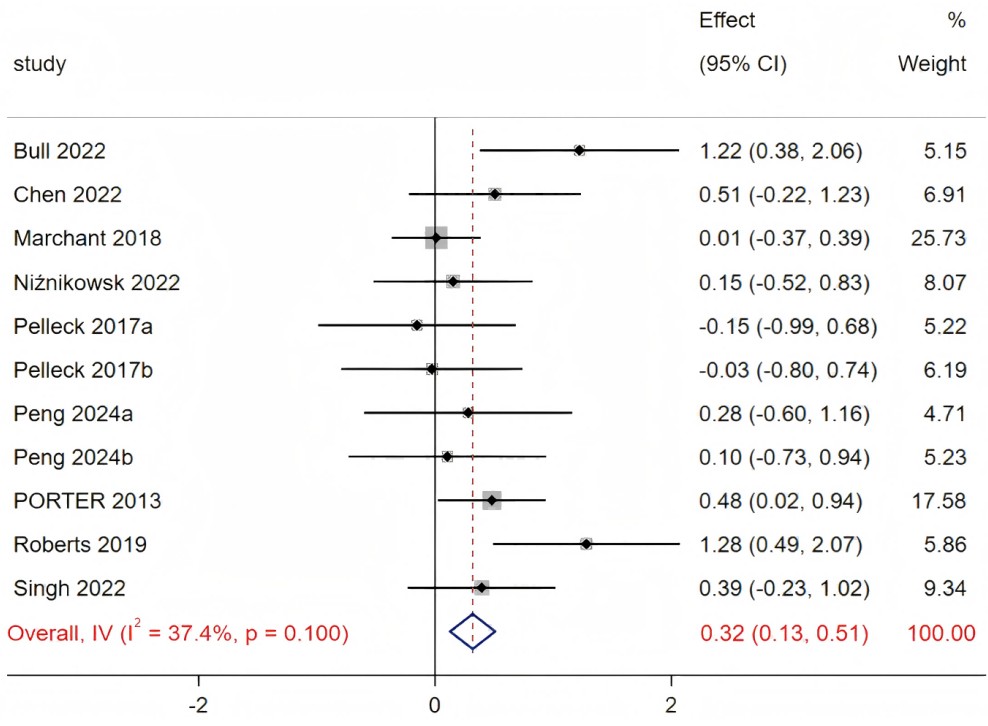

**Figure 4** **Forest plot for the effect sizes of the proximal external focus compared to the internal focus.** Sources: *Bull et al., 2023*; *Chen et al., 2022*; *Porter et al., 2013*; *Pelleck & Passmore, 2017*; *Roberts & Lawrence, 2019*; *Peng et al., 2024*; *Nagano, Hata & Nagano, 2020*; *Marchant et al., 2018*; *Singh et al., 2022*; *Singh & Wulf, 2020*; *Niznikowski et al., 2022*.

**Table 2** **Subgroup analysis of skill level, study design and skill type.**

| Variables | Subgroup category | Distal *vs* Proximal | | | | | Distal *vs* Internal | | | | | Proximal *vs* Internal | | | | |
|---|---|---|---|---|---|---|---|---|---|---|---|---|---|---|---|---|
| | | Studies | SMD | P | 95% CI | Heterogeneity testing | Studies | SMD | P | 95% CI | Heterogeneity testing | Studies | SMD | P | 95% CI | Heterogeneity testing |
| | | | | | | $I^2$ | P | | | | | $I^2$ | P | | | | | $I^2$ | P |
| Skill level | Experienced | 8 | 0.5** | <0.001 | 0.26 to 0.73 | 39% | 0.12 | 6 | 0.94** | 0.002 | 0.64 to 1.23 | 83% | <0.001 | 6 | 0.9* | 0.03 | 0.11 to 1.68 | 84% | 0.003 |
| | Novice | 7 | 0.1 | 0.36 | −0.12 to 0.32 | 41% | 0.12 | 5 | 0.13 | 0.34 | −0.14 to 0.4 | 5% | 0.38 | 5 | 0.12 | 0.43 | −0.15 to 0.39 | 0% | 0.38 |
| Study design | Within-subject | 16 | 0.33** | <0.001 | 0.15 to 0.5 | 24% | 0.18 | 9 | 0.44* | 0.002 | 0.23 to 0.65 | 67% | <0.001 | 9 | 0.38* | 0.01 | 0.08 to 0.68 | 45% | 0.07 |
| | Between-subject | 4 | 0.07 | 0.9 | −1.05 to 1.19 | 86% | <0.001 | 3 | 1.38 | 0.17 | −0.57 to 3.33 | 93% | <0.001 | 3 | 1.68 | 0.08 | −0.21 to 3.57 | 93% | <0.001 |
| Skill type | Open Skill | 6 | 0.59** | 0.008 | 0.25 to 0.93 | 4% | 0.37 | 3 | 1.15** | <0.001 | 0.53 to 1.77 | 50% | 0.13 | 3 | 0.52 | 0.07 | −0.04 to 1.07 | 46% | 0.16 |
| | Closed Skill | 14 | 0.2* | 0.01 | 0.04 to 0.35 | 48% | 0.02 | 9 | 0.4* | 0.03 | −0.05 to 0.84 | 75% | <0.001 | 9 | 0.3* | 0.04 | 0.1 to 0.5 | 78% | <0.001 |

**Notes.**
*$p < 0.05$.
**$p < 0.01$.

In the analysis of external *versus* internal focus, both distal (SMD = 0.94, 95% CI [0.64–1.23], $p < 0.001$) and proximal (SMD = 0.9, 95% CI [0.11–1.68], $p = 0.03$) external focus showed statistically significant improvements in motor performance among experienced participants compared to internal focus. However, no significant effects were found in the motor performance of novice participants ($p > 0.05$). With regard to study design,
significant effects of the intervention on motor performance were observed for both distal (SMD = 0.44, 95% CI [0.23–0.65], $p < 0.01$) and proximal (SMD = 0.38, 95% CI [0.08–0.68], $p = 0.01$) external focus within the within-group experimental design. No significant effects of external attentional focus were observed in the between-group experimental design ($p > 0.05$). In terms of skill type, both distal (SMD = 0.4, 95% CI [−0.05–0.84], $p < 0.05$) and proximal (SMD = 0.3, 95% CI [0.1–0.5], $p < 0.05$) external attentional foci were found to significantly enhance motor performance in closed skill. However, in open skills, the superior effects on sports performance were only reported in the distal focus group (SMD = 1.15, 95% CI [0.53–1.77], $p < 0.01$), while no significant advantage was found in the proximal external focus group ($p > 0.05$).

### Sensitivity analysis

To evaluate the stability of the meta-analysis results, a systematic sensitivity analysis was conducted by excluding each study individually. Following the excluding of individual studies, the $I^2$ values for the distal-proximal (47%–59%), distal-internal (56%–71%), and proximal-internal (34%–61%) comparisons showed that the $p$-values remained statistically significant. These finding suggested that the overall results of this meta-analysis are robust.

## DISCUSSIONS

The current meta-analysis included 20 studies investigating the effects of attention focus distance on motor performance. The results indicated that external attentional focus significantly improved motor performance compared to internal attentional focus, and the effect of improvement was influenced by spatial distance (*Ryuh et al., 2025*). As the spatial distance of the external focus increased, the advantage of distal external focus became more pronounced. Notably, this advantage is moderated by participants' skill levels. Subgroup analyses revealed that, compared to internal focus, external focus (including both distal and proximal) improved motor performance in experienced participants, while no significant improvements were observed in novice participants. These results differ from those of earlier meta-analyses. A previous meta-analysis indicated that the beneficial effect of a distal external focus is not influenced by age, health status, or skill level, suggesting that it can also improve performance in novices (*Chua et al., 2021*). However, the current meta-analysis does not clearly identify this advantage. Furthermore, the study only included 10 articles on the distance effect and did not directly compare distal external focus with internal focus. Another meta-analysis revealed that distal external focus significantly enhances novice performance compared to internal focus, but no significant difference was found when compared to proximal external focus (*Nicklas et al., 2022*). However, this study included only six between-group design studies and had a broad prediction interval, which limits the generalizability of the results across tasks and skill levels. Building on previous research, the current study specifically investigates the distance effect of external focus, incorporates a larger sample of studies, and considers both within-group and between-group designs, thus providing supplementary evidence for a more comprehensive understanding of the field.

Consistent with the findings of *Wulf et al. (2010)*, external focus demonstrated a significant advantage over internal focus in within-group designs. However, no such effect was detected in between-group designs. The discrepancy may arise from the methodological differences between within-group and between-group experimental designs. Specifically, in within-group designs, participants experience multiple conditions in a sequence, which could introduce carryover or sequence effects that influence subsequent performance. This means that the benefit of external focus might accumulate over time, making it more likely to show significant effects within the same group. Conversely, in between-group designs, each group is exposed to only one condition, and there is no opportunity for the effects of one condition to influence another. Moreover, the smaller sample size and moderate to high heterogeneity in between-group studies may reduce the statistical power to detect a significant effect. Importantly, external focus improves motor performance in both open and closed skills, and the effect of distal external focus is more significant than that of proximal external focus. It signifies that the distance effect of external focus was not influenced by skill type.

The results of this study do not fully support the constrained action hypothesis, theoretically. In the experienced group, the advantage of external focus increased with distance, whereas the performance of the novice group showed no significant differences across various focus distances. According to the constrained action hypothesis, the distance effect in the experienced group occurs because distal external focus reduces attentional resource consumption, promotes automated action control, and enhances motor performance (*Wulf & Lewthwaite, 2016*). Furthermore, it has been shown that distal external focus increases activation in the primary somatosensory cortex, motor cortex, and insular cortex, which may be associated with enhanced performance (*Favre-Bulle et al., 2024*). However, the results of the current meta-analysis are inconsistent with the predictions of the constrained action hypothesis. The current results indicate that distal external focus benefits only experienced participants.

The action-goal hierarchy hypothesis proposed by Stoate and Wulf may provide insight into this inconsistency. This hypothesis categorizes goals into different levels and suggests that hierarchical goals influence performance and learning outcomes based on participants' skill levels (*Stoate & Wulf, 2011*). For instance, experienced individuals typically focus on higher-level goals, such as "aiming at the center of the target" or "completing a jump over a specified distance". As a result, a distal focus is more beneficial for enhancing motor performance in experienced individuals. The subgroup analysis for experienced individuals supports this prediction, indicating that they have partially automated their motor skills. In this case, distal external focus helps direct attention toward higher-level goals. Research suggests that this attentional focus enhances neural pathways associated with the goal, optimizes the sensorimotor integration process, and helps the body adapt more quickly to task demands (*Chua et al., 2019*). In contrast, according to the action-goal hierarchy hypothesis, novices in the early stages of skill learning focus more on lower-level goals, such as "focusing on the swing trajectory of the club" or "concentrating on the acceleration of the ball". Therefore, proximal external focus may be more effective in improving novice performance. However, subgroup analyses did not support this prediction, as there was no

evidence that novices performed significantly better under the proximal external condition. A possible reason is that novices often struggle to distinguish their own body movements from task goals, making it difficult for them to benefit from different attention distances during task execution (*Freudenheim et al., 2010*; *Sherwood, Lohse & Healy, 2020*).

The current meta-analysis is not entirely consistent with the predictions of the action-goal hierarchy hypothesis. One potential explanation for these findings in the novice subgroup is that, during the early stages of skill acquisition, novice motor performance primarily depends on mastering fundamental movement patterns rather than on modulating attentional focus (*Wulf, Shea & Lewthwaite, 2010*). According to Fitts and Posner's three-stage model, the acquisition of motor skills progresses through the cognitive, associative, and autonomous stages (*Salehi, Tahmasebi & Talebrokni, 2021*). During the cognitive stage, novices primarily focus on mastering fundamental skills, fine-tuning motor movements, and making adjustments based on feedback (*Kee, 2019*). This stage requires substantial cognitive investment from novices as they strive to comprehend and coordinate multiple aspects of fundamental movements (*Kee, 2019*). Excessive cognitive load, resulting from this demanding process, can impede novices' ability to discern subtle variations in attentional focus, such as focusing on finger movements (internal) or focusing on the keys (external) when playing the piano. Consequently, altering attentional focus may not yield substantial enhancements in novice motor performance. Moreover, only six of the included studies compared the effects of external and internal focus at different distances on novices. The limited sample size of these studies may have biased the findings, suggesting that future research should further explore the effects of different attentional focus distances on novices.

A limitation of this study is that significant heterogeneity was reported in the heterogeneity test, although we endeavored to address this issue through subgroup analysis and sensitivity analysis, however, it remains important to treat this result with caution. The $I^2$ statistic from the subgroup analyses suggests that skill level may contribute to heterogeneity; however, inconsistency in the categorization of skill level ratings presents another limitation of this study. For instance, *Banks et al. (2020)* classified national-level athletes and international competitors as experienced groups, while *Bell & Hardy (2009)* defined skill level by years of play. This inconsistency made it challenging to accurately assess participants' expertise and compare them across studies. Additionally, a limitation of the included studies was the lack of standardized criteria for guideline development. In the two standing long jump studies included, the guidelines for distal focus were "Jump as close to the cone as possible" and "Focus on the +20 cm target line" (*Porter et al., 2013*; *Nagano, Hata & Nagano, 2020*). As *Wulf (2013)* noted in a review, focus of attention instructions should be as consistent as possible regarding the content and quantity of information presented to the performer. Therefore, when formulating instructions for concentration, different researchers should strive to maintain consistency in the amount and content of information. Finally, a major limitation lies in the interpretation of the results of subgroups of studies designed between novices and subjects. The small sample size of novices in the included randomized controlled trials and the inconsistent definition of novice (*e.g.,* those with no exposure to the sport at all *versus* those with no professional training were

considered novices) may reduce statistical efficacy and affect the comparability of results. In addition, the sample sizes of studies with a between-subjects design were similarly limited and more susceptible to confounding by individual differences, further increasing the difficulty of interpreting subgroup results.

Future research needs to investigate more closely the effects of distance of attentional focus on novice motor performance and learning, as well as the potential influence of between-subject design on distance effects. Furthermore, the effects of attentional focus on form-based skills such as dance and gymnastics remain unclear. Given the paucity of current research on such skills, it may be worthwhile for future research to investigate the distance effects of attentional focus in motor skills of different forms or complexity, thus further refining the use of attentional focus in instructional practice.

## CONCLUSIONS

The presented meta-analysis indicates that external focus of attention has been shown to effectively enhance experienced individuals' performance, with greater improvements observed as the focus distance increases. However, there is no evidence that novices can benefit from external focus at any distance. A key contribution of the present study is to demonstrate for the first time through quantitative research that the benefits of distal attentional focus are consistent for both open and closed motor skills. Furthermore, a within-subject design was more likely to reveal the benefits of distal attentional focus on motor performance than a between-subject design.

## ACKNOWLEDGEMENTS

During the preparation of this work, the authors used ChatGPT 4.0 to improve language and readability. After using the service, we reviewed and edited the content as necessary and take full responsibility for the content of the publication.

### Funding

This research was funded by the Natural Science Foundation of the Jiangsu Higher Education Institutions of China, grant number 23KJB320023. The funders had no role in study design, data collection and analysis, decision to publish, or preparation of the manuscript.

### Grant Disclosures

The following grant information was disclosed by the authors:
Natural Science Foundation of the Jiangsu Higher Education Institutions of China: 23KJB320023.

### Competing Interests

The authors declare there are no competing interests.

## Author Contributions

- Le Zang performed the experiments, analyzed the data, prepared figures and/or tables, and approved the final draft.
- Wei Guo conceived and designed the experiments, analyzed the data, prepared figures and/or tables, authored or reviewed drafts of the article, and approved the final draft.
- Biye Wang conceived and designed the experiments, analyzed the data, authored or reviewed drafts of the article, and approved the final draft.

## Data Availability

The raw measurements are available in the Supplementary Files.

## Supplemental Information

Supplemental information for this article can be found online at http://dx.doi.org/10.7717/peerj.20012#supplemental-information.

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
