# Peer review of "The farther, the better? The effect of attentional focus distance on motor performance: a systematic review and meta-analysis"

_PeerJ, doi:10.7717/peerj.20012_

## Round 0.1 · original submission · Major Revisions

·

Basic reporting

Lines 41–42: The phrase “focusing on bodily sensation” may not fully capture the concept of internal focus, as it involves not only sensations but also the movement itself. For example, a common internal focus cue for jumping tasks (e.g., standing long jump, vertical jump) is “focus on extending the knees as fast as possible,” which refers to a specific movement, not just a sensation. I recommend replacing this phrase with “focusing on body movement,” which better reflects the broader scope of internal focus. Similarly, the description “focusing on the goal or external environment” does not comprehensively define external focus. The external environment may include regulatory conditions that directly influence performance as well as non-regulatory conditions. Moreover, performers can still focus on their goal while attending to internal cues. If performers are told to focus on the internal cue, then to focus on an internal cue, then it becomes the means to achieve their action goal. I suggest adopting a more widely accepted definition, such as “focusing on an object, movement effector, or target ” which more accurately reflects the established understanding of external focus.

Line #54: I would avoid using the word “excessive,” as it makes the interpretation of internal focus ambiguous. It is unclear whether the issue lies with internal focus itself or with its excessive application. Also, how much focus would be considered excessive?

Line #70-74: I don't think this sentence is accurate. The Constrained Action Hypothesis itself does not distinguish between proximal and distal external focus. It posits that focusing on an internal cue leads to conscious movement control, which interferes with automatic, or subconscious, processes. In contrast, an external focus promotes more automatic processing of movement control. However, the hypothesis never mentioned that distal external focus is more effective or supported, while proximal external focus is not. Distal effects of motor learning are established by follow-up researchers.

Line #78-79: Also not an accurate sentence. The hypothesis itself did not mention skill level.

Line #85: Typo – it should be 'a significant,' not 'an significant.' Also, the study never said that internal focus is advantageous for novice golf putting performance. Rather, it found that distal external focus may be detrimental for novices when performing complex motor skills for which they have not yet developed a motor program. According to the study, proximal external focus was proven to be the most effective cue among internal, proximal external, distal external, and control condition.

Experimental design

Based on what I read in lines #124–125, this study incorporated the classification of open vs. closed skills into the existing two review studies, but there are a total three subgroup analyses including open vs. closed, skill level (experienced vs. novice) and study design (within-subject vs. between-subject). Please mention the other two here.

Validity of the findings

As noted in line # 117-120, a recent meta analytic systematic review paper by Chua et al. (2021) has already well-established the superiority of an external focus of attention (whether proximal or distal) on motor performance and learning, regardless of performers’ age, health status, or skill level. Their work further addresses distal effects of attentional focus though and this study being reviewed indicated the same results. My concern here is that the study did not justify the meaningfulness of their replication. However, your study did make some meaningful contributions by documenting that distal focus enhances both open and closed skills. Additionally, the within-subject design (which may better reflect motor performance measures) showed a more notable benefit from distal effect of attentional focus compared to the between-subject design (which may represent motor learning measures). I suggest highlighting this novelty of the study in the discussion or conclusion section to emphasize the unique contributions of your study.

Additional comments

Overall, this is well-analyzed study that has a promising potential to contribute to the current body of knowledge. However, I suggest working on clearly addressing definitions or notions in the introduction section such as Constrained Action Hypothesis as commented above.

Also, moving on, there are emerging evidence that form-based skills, such as dance pirouette, gymnastics routine, dance, or figure skating, there is no difference in motor performance and learning induced by attentional focus. Since this is "form-based" internal focus could also be an "outcome-producing" focus cue. Historically, researchers have been primarily selecting a projection skill with evident target, and that skill selection might lead external, especially distal, focus to being superior due to higher relevance. You can related this to the Common Coding Theory. You may want to consider this skill category into account for your subsequent review study. Keep it up!

Some related references:
Andrade, C. M., Souza, T. R. D., Mazoni, A. F., Andrade, A. G. D., & Vaz, D. V. (2020). Internal and imagined external foci of attention do not influence pirouette performance in ballet dancers. Research Quarterly for Exercise and Sport, 91(4), 682-691.
Chua, T. X., Sproule, J., & Timmons, W. (2018). Effect of skilled dancers' focus of attention on pirouette performance. Journal of Dance Medicine & Science, 22(3), 148-159.
Teixeira da Silva, M., Thofehrn Lessa, H., & Chiviacowsky, S. (2017). External focus of attention enhances children's learning of a classical ballet pirouette. Journal of Dance Medicine & Science, 21(4), 179-184.

Reviewer 2 ·

Basic reporting

This meta-analysis study examines how the location of attentional focus affects motor performance. Three conditions were considered: internal focus (within the participant), proximal external focus (near the participant), and distal external focus (far from the participant). Consistent with previous studies, including review papers, the meta-analysis results showed that performance was highest in the order of distal external, proximal external, and internal focus. Furthermore, the authors investigated whether these differences could be attributed to task proficiency (novice vs experienced), experimental design variations (within-subject vs between-group), and differences in motor tasks (close vs open) using a subgroup analysis. The analysis followed standard procedures, and I did not find major methodological issues. The inclusion criteria for the studies in the meta-analysis are clearly stated and ensure reproducibility. As a non-native speaker, I will refrain from making judgments on the quality of the English.
Below, I list several points of concern and areas for revision, along with minor corrections that should be addressed.

Experimental design

The paper attempts three subgroup analyses. However, the necessity of analyzing task type (open vs. closed) is barely discussed. Since this analysis is crucial for defining the importance and necessity of the study, explicitly justify its inclusion. In addition, regarding the necessity of examining experiment design (within vs. between groups), it is briefly mentioned around line 110, where the authors state that "each design type carries distinct biases." While I fully agree that different designs introduce different biases, the paper should clearly define these distinctions. A precise definition here will strengthen the validity of the interpretations in the Discussion section.

Validity of the findings

The statistical values reported in the main text for the subgroup analysis appear to differ from those in Table 2. For instance, where are the statistics for the novice vs. experienced comparison (lines 304-305) listed in the table? Additionally, the corresponding confidence intervals (CIs) seem to be missing. Until these inconsistencies are resolved, it is difficult to determine whether the results truly support the authors' claims. In addition, please ensure consistency in significant digits between the main text and tables. In Table 1, what does the "P" value represent? It seems different from the p-values described in the text. The descriptions of the table should be more detailed.

Additional comments

Minor Comments
Line 93: What is the OPTIMAL theory? Please provide a brief explanation.
Line 180: The reference "(8)" cannot be found. While the authors claim there are ten key aspects, I could only identify nine. Please verify.
Table 1: Change "0:05" to "0.01".
Line 214: Capitalize "Sensitivity".
Lines 217-219: Provide more details about the analysis used to check publication bias, including a nonparametric trim-and-fill analysis. This will help readers replicate the methodology.
Lines 242-243: The authors mention that eight studies focus on experienced individuals and seven on novices. What about the remaining five studies? Looking at Table 1, it appears this information might be missing from the paper. If so, explicitly state this in the text for clarity.
Line 310: Remove the unnecessary comma.
Lines 323-325: If claiming that an effect was observed, report the statistical values supporting this claim.

Reviewer 3 ·

Basic reporting

Overall, the manuscript is clearly written and well-organized and easy to follow.

The introduction uses key theories like the Constrained Action Hypothesis and the OPTIMAL Theory and connects them well to the study’s aims. However there is a lot of repetitive or overly detailed discussion, especially when revisiting the same theoretical concepts.

The authors follow PRISMA guidelines, the protocol is registered with PROSPERO, and the use of the PICOS framework is clearly outlined.

The figures and tables are relevant but hard to interpret at first glance. It is recommended that clearer labels, more spacing, or highlighting the key results.

Raw data are included in the supplementary files, and the manuscript as a whole is self-contained. The results are clearly tied back to the original hypotheses, and the inclusion of subgroup and sensitivity analyses adds helpful depth and robustness to the findings.

Experimental design

A few areas could be clarified or expanded to strengthen the methodological design. For example, while the inclusion and exclusion criteria are thorough, more detail on how discrepancies during study screening and data extraction were resolved would be helpful, particularly whether inter-rater reliability was assessed or documented. Although the use of both within- and between-subject designs is discussed in the results, it would be helpful to more explain why both were included and how potential differences in design were accounted for in the analysis.

The risk-of-bias assessment is carefully executed using the Cochrane tool, but the discussion of bias could be brought forward earlier in the methods section to prepare readers for how these factors may affect interpretation.

Lastly, while the methods section is detailed overall, it is not clear how composite effect sizes were calculated when multiple tasks were used in a single study. Make sure methods are described in a replicable manner.

Validity of the findings

One area for improvement would be to more explicitly address the limitations of subgroup findings involving novices and between-subject designs. While these are acknowledged, the discussion could benefit from a clearer explanation of why some comparisons are underpowered or more difficult to interpret. The smaller sample sizes in novice-related studies and the inconsistent definitions of “novice” across the included RCTs introduce interpretive challenges that should be more clearly flagged for readers.

It is recommended to directly state where replication is needed, and how future studies could help refine best practices for attentional focus instruction.

---

## Round 0.2 · Minor Revisions

·

Basic reporting

The line numbers as follows align with a Word document with track changes.

#80: While McNevin et al. (2003) were among the first to empirically demonstrate this within a specific experimental context, the phrasing "first to propose" may overstate their role as there are many former research papers stating the benefit of external focus on motor performance and learning such as Wulf & Prinz (2001). Rephrase it without overstatement.

"...more effectively dissociating the outcome..." I am sure what this part means. The listed references this part just indicating that distal external focus is better than proximal, or internal focus. Does this statement mean that they separate movement execution and movement outcome? However, I don't think they have separated this as perception-action coupling is very important. For example, recent work (not published yet) from the leading scholars in the area of attentional focus found that external focus is better when there is a bidirectional relationship between movement production and response outcome. That said, performers have to connect external outcome and their movement production to make the external focus beneficial. Thus, "dissociating" is a bit dangerous explanation.

#88: I would suggest adding Chua et al.'s (2021) meta analytic paper here: Chua, L. K., Jimenez-Diaz, J., Lewthwaite, R., Kim, T., & Wulf, G. (2021). Superiority of external attentional focus for motor performance and learning: Systematic reviews and meta-analyses. Psychological bulletin, 147(6), 618.

#132-137: Be cautious here. "Proximal external focus" (e.g., focusing on the club) is still external but closer to the body than focusing on the target. However, this is conceptually distinct from internal focus, which refers to focusing on body parts or movement mechanics (e.g., wrist movement). The literature (including Lohse et al.) more often contrasts external (proximal or distal) with internal focus, not proximal vs distal in relation to choking.

#150: "RANGANATHAN, LEE & KRISHNAN, 2022" please correct this citation.

#491: Correct the grammar please.

Experimental design

NA

Validity of the findings

#456. It is important to clarify that a proximal focus still falls under the category of external focus—distinguished from distal external focus by its closer spatial relation to the body or action. However, the manuscript seems to treat proximal focus as if it were an internal focus, which reflects a conceptual misunderstanding that should be addressed for clarity and theoretical consistency. Read this to clarify: Ryuh, Y., Geschwender, C., Kim, J., & Becker, K. (2024). The Distance Effect in Focus of Attention: Spatial or Temporal Distance?. Research Quarterly for Exercise and Sport, 1-6.

Throughout the manuscript, the terms “proximal focus” and “distal external focus” are used inconsistently. I recommend standardizing the terminology, preferably revising “proximal focus” to “proximal external focus.” For example, in lines #517–521, the authors categorize proximal and internal focus together, which is conceptually inaccurate. In the literature, proximal and distal are typically grouped under external focus, while internal focus is considered a separate category. Given that the authors frequently reference the constrained action hypothesis which explicitly supports external focus (both proximal and distal), this repeated conceptual errors raises concerns about their understanding of theoretical foundation. In my opinion, there is a need for deeper understanding about attentional focus literature.

Reviewer 2 ·

Basic reporting

I have checked the response letter and re-submitted manuscript, and found the authors adequately revised the manuscript along with my comments in the previous review round. Thus, now, I have nothing to aadd, and would finalize my review process.

Experimental design

Nothing particularly to add now.

Validity of the findings

Nothing particularly to add now.

---

## Round 0.3 · accepted · Accept

Overall, two reviewers are satisfied with your revision. However, I noticed that the first reviewer provides two small suggestions. Please check it and make your manuscript perfect.

·

Basic reporting

(PDF file)
Line #71 & #128: incorrect citation format. Please put the authors' last names only.

#75: "Distant" external focus is inconsistent terminology, as other parts of the manuscript, as well as other literature, used "distal" external focus instead.

Overall, please double-check the APA style citation.

Experimental design

-

Validity of the findings

-

Additional comments

-

Reviewer 2 ·

Basic reporting

I now have nothing that should be added to the present manuscript.

Experimental design

Nothing particularly now.

Validity of the findings

Nothing particularly.